# Residual Triplet Attention Network for Single-Image Super-Resolution

**Feng Huang, Zhifeng Wang, Jing Wu, Ying Shen *** and **Liqiong Chen**

School of Mechanical Engineering and Automation, Fuzhou University, Fuzhou 350108, China;
huangf@fzu.edu.cn (F.H.); N190227101@fzu.edu.cn (Z.W.); wujing@fzu.edu.cn (J.W.);
liqiongchen@whu.edu.cn (L.C.)
* Correspondence: yshen@fzu.edu.cn

**Abstract:** Single-image super-resolution (SISR) techniques have been developed rapidly with the remarkable progress of convolutional neural networks (CNNs). The previous CNNs-based SISR techniques mainly focus on the network design while ignoring the interactions and interdependencies between different dimensions of the features in the middle layers, consequently hindering the powerful learning ability of CNNs. In order to address this problem effectively, a residual triplet attention network (RTAN) for efficient interactions of the feature information is proposed. Specifically, we develop an innovative multiple-nested residual group (MNRG) structure to improve the learning ability for extracting the high-frequency information and train a deeper and more stable network. Furthermore, we present a novel lightweight residual triplet attention module (RTAM) to obtain the cross-dimensional attention weights of the features. The RTAM combines two cross-dimensional interaction blocks (CDIBs) and one spatial attention block (SAB) base on the residual module. Therefore, the RTAM is not only capable of capturing the cross-dimensional interactions and interdependencies of the features, but also utilizing the spatial information of the features. The simulation results and analysis show the superiority of the proposed RTAN over the state-of-the-art SISR networks in terms of both evaluation metrics and visual results.

**Keywords:** image super-resolution; attention mechanism; convolutional neural networks; deep learning

## 1. Introduction

Single-image super-resolution (SISR) is a well-known technique in computer vision which is used to reconstruct degraded low-resolution (LR) images and convert them into high-resolution (HR) images. However, this is an ill-posed problem, since there are numerous HR outputs for one LR image. In the literature, there are various efficient methods presented to address this issue, such as the interpolation-based method [1], sparse representation methods [2,3], and learning-based methods [4–13].

With the gradual maturity of deep-learning technologies, image super-resolution has also made a significant breakthrough. The convolutional neural networks (CNNs) possess powerful learning capabilities in different application scenarios [14,15], which enable us to complete end-to-end training of image super-resolution. Dong et al. [5] proposed the Super-Resolution Convolutional Neural Network (SRCNN) which consists of three-layer CNN and learns the non-linear mapping from LR image to its corresponding HR output. Since then, a variety of CNNs-based techniques [16–21] were applied to tackle SR tasks by designing deeper or wider networks.

Although impressive results have been achieved in SISR, the existing methods based on CNNs still face several challenges. First, very deep, and very wide networks are usually accompanied by large number of parameters, which is not beneficial to practical applications. Second, most super-resolution models are unable to exploit the advantages

of CNNs to further extract the information in the LR image. Third, stacking the complex modules usually ignores the interdependence between different dimensions of the features.

To address these problems effectively, we propose a residual triplet attention network (RTAN), which considers the cross-dimensional interdependence and interaction of the features for more powerful feature representations. Specifically, inspired by the effect of the residual structure which contains the convolution layer, the RELU function and the shortcut connection [6,11,17,22], we propose the multiple nested residual group (MNRG) structure to reduce model degradation and reuse more informative LR features. In MNRG, we adopt the global shortcut connection (GSC) which serves as the first-layer structure to complete the rough learning of LR image. The local shortcut connection (LSC) and the third-layer residual module are further employed to alleviate the training difficulty causing by the network depth and learn abstract residual features. Moreover, motivated by the attention mechanism demonstrated in [23–25], a residual triplet attention module (RTAM) is proposed to improve the interactions and interdependencies of the deep residual features. The RTAM mainly contains two cross-dimensional interaction blocks (CDIBs) and one spatial attention block (SAB). The CDIB has the ability to capture the interaction information between the channel dimension C and spatial dimension (W and H) of the features. This mechanism enables the proposed network to acquire more blending cross-dimensional feature information. Meanwhile, the SAB further extracts the spatial information and helps the network to discriminate the spatial locations of the features. The RTAM explores the interdependencies and interactions across the dimensions of the features without introducing too many parameters. By stacking the RTAM, we further bypass most of the low-frequency part in the input LR and fully exploit the feature information from the intermediate layers of the network.

The main contributions of the proposed RTAN are summarized below.

(1)   A residual triplet attention network (RTAN) is proposed to make full use the advantages of CNNs and recover clearer and more accurate image details. The comprehensive simulations demonstrate the effectiveness of the proposed RTAN over other chosen SISR models in terms of both evaluation metrics and visual results.

(2)   We design the multiple nested residual group (MNRG) structure which reuses more LR features and diminishes the training difficulty of the network.

(3)   A residual triplet attention module (RTAM) is proposed to compute the cross-dimensional attention weights by considering the interdependencies and interactions of the features. The RTAM uses the inherent information between the spatial dimension and channel dimension of the features in the intermediate layers, thus achieving sharper SR results and further applying to actual scenes.

This paper is organized as follows. In Section 2, related works on the image-super-resolution and attention mechanism are introduced. In Section 3, the proposed methods are presented. In Section 4, some discussions on RCAN and RTAN are provided. In Section 5, the experimental results and analysis on different benchmark datasets and the ablation study on the proposed network are given. Model complexity comparisons are also included. In Section 6, the conclusions of the paper are drawn.

## 2. Related Work

In the past few decades, the image super-resolution has made remarkable progress in computer vision. The researchers have proposed numerous of techniques to address the ill-posed issue in single image SR. There are two categories of the proposed solutions, namely, traditional methods and CNN-based techniques. Owing to the effective learning ability of the CNNs, SISR has been developed rapidly. In this section, we firstly present the related techniques considering the SISR based on CNNs. Then, we briefly discuss the attention mechanism, which inspires our work.

### 2.1. CNNs-Based Single Image Super-Resolution Network

Dong et al. [5] applied the CNN to present the pioneering work in image super-resolution. The SRCNN that comprises a three-layer convolutional neural network was firstly proposed by the authors to perform end-to-end learning of image super-resolution. As compared with the traditional solutions, this method shows prominent performance. Kim et al. applied the residual learning strategy to image SR, proposed VDSR by increasing the network depth [12] and DRCN which constructs a very deep recursive layer [26]. Tim et al. presented DRRN by using the recursive learning [27] to control the model parameters and adopted the memory blocks in MemNet [28]. These SR networks first perform interpolation on LR input images to get coarse HR images with the desired size before the feature extraction layer and reduce the learning difficulty, while having relatively large memory and computational overhead. In order to address this issue, Shi et al. [29] designed the sub-pixel layer. This layer is a learnable up-sampling layer and performs the convolution and reshaping operations. Inspired by the technique of sub-pixel layer, an increasing number of excellent SR models were proposed. Lim et al. [17] developed the EDSR which significantly improved SR performance. The authors removed the batch normalization in the original residual blocks and the modified residual module consists of two convolution layers, the ReLU function and the shortcut connection. Other works, such as ESRGAN [30], MemNet [28], and RDN [16], utilize the dense connections by using all the hierarchical features of the convolutional layers. Some recent networks focus on handing the trade-off between performance of the SR and memory consumption. For example, Lai et al. [18] presented LapSRN by reconstructing the sub-band residuals of HR image progressively. Ahn et al. [31] proposed CARN, which uses the group convolution to make the image SR network lightweight and efficient. A few networks further explore the feature correlations in spatial or channel dimensions, such as NLRN [32], and RCAN [6]. Moreover, several works aim to exploit more efficient networks to improve SR performance, such as IMDN [33], OISK [34], NDRCN [35], SMSR [36], FALSR [37] and ACNet [38]. Recently, some researchers adopted the graph convolution network [39] and proposed IGNN [40].

Most of the forementioned techniques only consider the design of architecture of the network to achieve better SR results by making the network deeper or wider. However, most of these methods do not consider the correlation between different dimensions of the features and do not fully utilize the advantages of CNNs, which are good at extracting the inherent features.

### 2.2. Attention Mechanism

The well-known attention mechanism is an effective means of biasing the distribution of available computing resources to the most useful part of the input signals [23]. The attention mechanism is widely applied in many tasks of computer vision, such as image classification [23,41], semantic segmentation [42], human posture estimation [43], and scene parsing [44]. Some tentative works have been proposed which achieve good performance in high-level computer tasks [45]. Wang et al. [41] exploited an effective bottom-up top-down attention mechanism for image classification. Hu et al. [23] proposed the block by squeezing and exciting to obtain the relationship between feature channels. The squeezing operation is completed by using global average pooling and the exciting operation is finished by adopting the MLP and sigmoid function. This technique improves the performance of the existing CNNs. More recently, Wang et al. [46] proposed a novel non-local block, which computes the response at a position as a weighted sum of the features at all possible positions in the input feature maps. Woo et al. [25] proposed a lightweight module (CBAM) with the channel and spatial attention mechanism.

Recently, some researchers have proposed attention-based models to improve the SR performance. Several works, such as RCAN [6], SAN [21], MCAN [47], A$^2$F [48], LatticeNet [49], and DIN [50], introduce the channel attention (CA) mechanism to SR which makes the network learn more useful features. To learn more discriminative features, some researchers utilize both the channel attention and spatial attention, such as

HRAN [51], MIRNet [52], CSFM [53], and BAM [54]. Additionally, SAN [21], NLRN [32], and RNAN [55] use the non-local attention to capture the long-term dependencies between pixels in the image. Mei et al. [56] proposed the CSNLN which combines the recurrent neural network with the cross-scale non-local attention to explore the cross-scale feature correlations. Liu et al. [57] proposed the enhanced spatial attention (ESA) which adopts a strided convolution with a larger stride followed by the max-pooling operation. This method can enlarge the receptive field effectively. Muqeet et al. [58] proposed the efficient MAFFSRN by modifying the ESA with the dilated convolutions to refine the features. Zhao et al. [59] introduced the pixel attention mechanism to SR and designed the effective PAN. Mei et al. [60] designed the pyramid attention network (PANet) to capture multi-scale feature correspondences. Huang et al. [61] proposed the DeFiAN to recover high-frequency details of the images by introducing a detail-fidelity attention mechanism. Wu et al. [62] proposed the multi-grained attention network (MGAN) by measuring the importance of every neuron in a multi-grained way. Chen et al. [63] proposed the attention dropout module in A2N to adjust the attention weights dynamically.

Although this method of computing the channel attention weight is proven to be effective, it results in a major loss of spatial information due to the global average pooling. Other methods usually bring huge computation overhead and complicated operations. In this work, a residual triplet attention network (RTAN) is proposed to exploit the internal relations across different dimensions of the features in an effective way.

## 3. Proposed Methods

### 3.1. Network Architecture

The overall architecture of RTAN comprises four parts, namely the shallow feature extraction module, the multiple nested residual deep feature extraction part, the upscale module and reconstruction section, as shown in Figure 1, where $I_{lr}$ and $I_{sr}$ represent the input and output of RTAN, respectively.

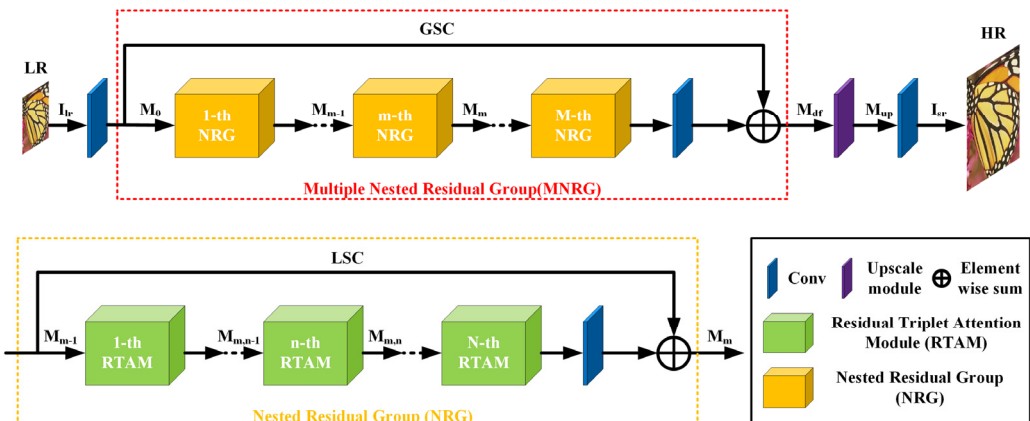

**Figure 1.** The overall architecture of the residual triplet attention network (RTAN).

A $3 \times 3$ convolutional layer of 64 filters is applied to extract the shallow features from the LR input.

$$M_0 = H_{SF}(I_{lr}), \tag{1}$$

where $H_{SF}(\cdot)$ denotes the function of feature extraction. $M_0$ denotes the input of MNRG structure to achieve the deeper feature extraction. Thus,

$$M_{df} = H_{MNRG}(M_0), \tag{2}$$

where $H_{MNRG}(\cdot)$ denotes the MNRG structure, which is stacked from series of nested residual groups (NRGs). The proposed MNRG structure not only enlarges the size of the receptive field, but also performs substantially by learning the local residuals of the input.

Then, the extracted deep feature $M_{df}$ is passed to the upscale module. The subsequent procedure is expressed as follows.

$$M_{up} = H_{UP}\left(M_{df}\right) = PS\left(W_{L-1} * M_{df} + b_{L-1}\right), \tag{3}$$

where $H_{UP}(\cdot)$ denotes the function of upscale module. $PS(\cdot)$ denotes the operation of periodic shuffling. $W_{L-1}$ and $b_{L-1}$ denote the weights and biases of the upscale module, respectively. The symbol "*" denotes the convolution product. $L$ denotes the number of RTAN layers. Please note that $M_{up}$ denotes the upscaled feature maps. As investigated in [29], we utilize the sub-pixel layer as the upscale module. The mathematical expression is as follows.

$$PS(T)_{x,y,c} = T_{\lfloor \frac{x}{r} \rfloor, \lfloor \frac{y}{r} \rfloor, c \cdot r \cdot \mathrm{mod}(y,r) + c \cdot \mathrm{mod}(x,r)}, \tag{4}$$

where $PS(\cdot)$ is a periodic shuffling operator which can rearrange the input features maps $H \times W \times r^2 \times C$ to the output with the shape of $C \times rH \times rW$, $r$ denotes the scale factor. $x, y$ denote the output pixel coordinates in the HR image space. $c$ denotes the number of channels. $\mathrm{mod}(y,r)$ and $\mathrm{mod}(x,r)$ denotes the different sub-pixel location. $T$ denotes the input tensor. The symbol $\lfloor \cdot \rfloor$ denotes the rounding-down operation.

Afterwards, we reconstruct the upscaled features $M_{up}$ via one $3 \times 3$ convolutional layer. The process of the final module is formulated as

$$I_{sr} = H_{RE}\left(M_{up}\right) = H_{RTAN}(I_{lr}), \tag{5}$$

where $H_{RE}(\cdot)$ and $H_{RTAN}(\cdot)$ denote the reconstruction layer and the function of *RTAN*, respectively.

Then, the *RTAN* will be optimized with a certain loss function. There are various loss functions used in previous SR works, such as L1 [5,12], L2 [17,18], perceptual and adversarial losses [11]. We train the proposed *RTAN* with the L1 loss function. Given the training datasets $\left\{I_{LR}^i, I_{HR}^i\right\}_{i=1}^N$ with $N$ LR images $I_{LR}$ and their *HR* counterparts $I_{HR}$, the aim of training *RTAN* is to minimize the mathematical expression as follows.

$$L(\theta) = \frac{1}{N}\sum_{i=1}^N \left\| H_{RTAN}\left(I_{LR}^i - I_{HR}^i\right) \right\|_1, \tag{6}$$

where $\theta$ denotes the parameter set of the proposed *RTAN*.

### 3.2. Multiple Nested Residual Group (MNRG) Structure

Here, we present the details regarding the proposed multiple nested residual group (MNRG) structure (see Figures 2 and 3), which consists of M nested residual group (NRG) structure with a global shortcut connection (GSC). Each NRG further contains N RTAMs and a local shortcut connection (LSC). Several works [17,22] demonstrate that stacking the residual blocks achieves accuracy gains due to increased depth. However, there is a higher training difficulty as the depth increases. Inspired by previous works presented in [6,11,17], we propose the NRG structure as the fundamental unit. The input and output feature maps $M_{m-1}$, $M_m$ of the m-th NRG satisfy the following expression.

$$M_m = F_{NRG,m}(M_{m-1}) = F_{NRG,m}(F_{NRG,m-1}(\cdots F_{NRG,1}(M_0)\cdots)), \tag{7}$$

where $F_{NRG,m}$ represents the function of m-th *NRG*.

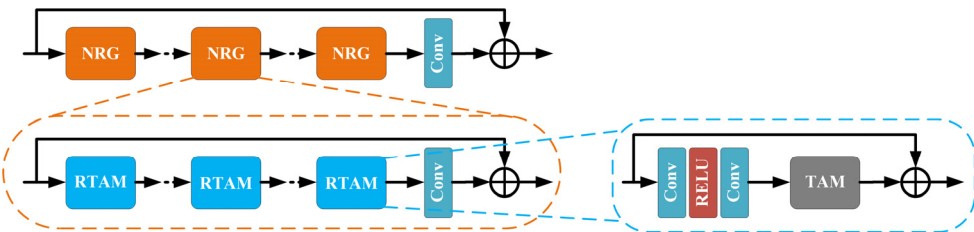

**Figure 2.** The framework of the proposed multiple nested residual group (MNRG) structure.

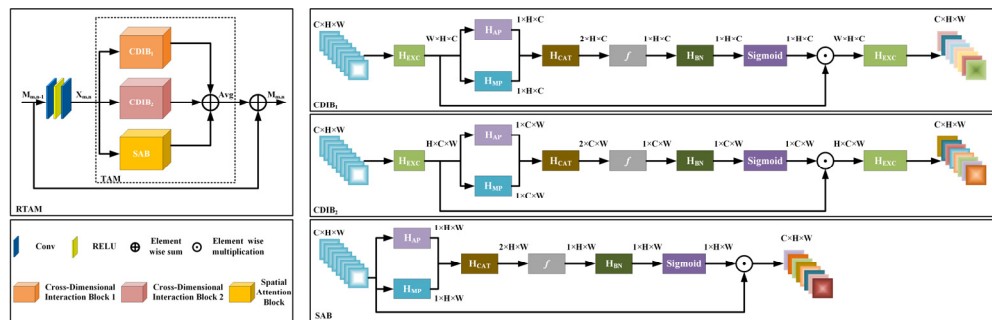

**Figure 3.** The framework of the proposed residual triplet attention module (RTAM).

Since the LR and HR image have a highly similar structure, we only need to learn the residual feature mapping to recover the losing high-frequency image information. As discussed in [12], we use global shortcut connection (GSC). Thus, the computation and the learning difficulty of the model are substantially reduced. This procedure is expressed as

$$M_{df} = M_0 + W_{GSC}M_M = M_0 + W_{GSC}(F_{NRG,m}(F_{NRG,m-1}(\cdots F_{NRG,1}(M_0)\cdots))), \quad (8)$$

where $M_{df}$ denotes the deep feature maps. $W_{GSC}$ represents the set of weight parameters for the convolutional layer at the end of the final *NRG* in the MNRG structure.

We further present the *RTAM* to learn more abstract features and reduce the training difficulty. Each *NRG* structure comprises N stacked *RTAMs*. The *n*-th *RTAM* in the *m*-th NRG is represented as

$$M_{m,n} = F_{RTAM,m,n}(M_{m,n-1}) = F_{RTAM,m,n}(F_{RTAM,m,n-1}(\cdots F_{RTAM,m,1}(M_{m-1})\cdots)), \quad (9)$$

where $M_{m,n-1}$ and $M_{m,n}$ represent the input and output feature maps of the *n*-th *RTAM* in the *m*-th *NRG*. $F_{RTAM,m,n}$ denotes the corresponding function.

Image information can be divided into high-frequency components (i.e., edges, texture, and other details.) and low-frequency components (i.e., flat area). Since the residual structure can learn the mapping from LR to HR, it bypasses plenty of redundant low-frequency parts in the input LR image. In order to further improve the learning ability of the representational features and ease the difficulty of training, the local shortcut connection (LSC) is further employed to generate the *m*-th NRG output via

$$M_m = M_{m-1} + W_m M_{m,N} = M_{m-1} + W_m F_{RTAM,m,N}(F_{RTAM,m,N-1}(\cdots F_{RTAM,m,1}(M_{m-1})\cdots)), \quad (10)$$

where $W_m$ represents the weight of the convolutional layer behind the final *RTAM*.

The GSC along with the LSC encourages the network to learn more residual information which contains more high-frequency features of the LR. In addition, they also alleviate the network degradation caused by the ever-increasing network depths.

### 3.3. Triplet Attention Module (TAM)

3.3.1. Cross-Dimensional Interaction Block (CDIB)

As discussed in [23–25], a CDIB is proposed to compute the attention weights by capturing the cross-dimensional interaction by using a two-branch module. The two branches help the CNNs to make better use of the existing feature information, and fully explore the correlation and dependence between the features. As presented in Figure 3, the input $X_i \in \mathbb{R}^{C \times W \times H}, i = 1, 2$ is sent to two CDIBs simultaneously. In $CDIB_1$, the height H and the channel C interact with each other. We exchange dimension C and dimension W of the input feature maps. Then, the exchanged input is denoted as

$$\widetilde{X}_1 = H_{EXC}(X_1), \tag{11}$$

where $H_{EXC}(\cdot)$ denotes the permutation operation, $\widetilde{X}_1 \in \mathbb{R}^{W \times H \times C}$.

We further reduce the dimension W of the $\widetilde{X}_1$ to 2 by concatenating the average pooled and max pooled feature maps across that dimension. We obtain the output $\widetilde{X}_1^* \in \mathbb{R}^{2 \times H \times C}$ as follows.

$$\widetilde{X}_1^* = H_{CAT}\left(H_{MP_{0d}}\left(\widetilde{X}_1\right), H_{AP_{0d}}\left(\widetilde{X}_1\right)\right), \tag{12}$$

where $H_{CAT}$ represents the concatenation of the given input sequences in the specified dimension. $H_{MP_{0d}}$ and $H_{AP_{0d}}$ represent the max-pooling and the average-pooling function in the zero-th dimension, respectively.

The pooling layer contributes to preserve the rich information of the original features while reducing the computations. Then, $\widetilde{X}_1^*$ is passed through one convolutional layer with the filter size $7 \times 7$ followed by a batch normalization layer.

$$\widetilde{X}_1^{**} = H_{BN}\left(f^{7 \times 7}\left(\widetilde{X}_1^*\right)\right), \tag{13}$$

where $f^{7 \times 7}$ denotes the convolutional layer with the filter size $7 \times 7$. $H_{BN}(\cdot)$ represents the batch normalization.

As discussed in [6,23,24], we use the sigmoid activation function $\delta$ as the gating mechanism to compute the resultant attention weight. The computed attention weight maps are subsequently used to rescale $\widetilde{X}_1$, and then dimension swapping was performed to restore the same dimensional order as the original input $X_1$. Now, we obtain the final output, i.e.,

$$\widetilde{X}_1^{***} = H_{EXC}\left(\delta\left(\widetilde{X}_1^{**}\right) \cdot \widetilde{X}_1\right), \tag{14}$$

Similar to $CDIB_1$, in $CDIB_2$, we create interactions between the channel dimension $C$ and the width dimension $W$. The input $X_2$ is also exchanged between dimension $C$ and dimension $H$. The exchanged feature maps are denoted as $\widetilde{X}_2 \in \mathbb{R}^{H \times C \times W}$. Then, $\widetilde{X}_2$ is input into the pooling layer. We further get the output $\widetilde{X}_2^* \in \mathbb{R}^{2 \times C \times W}$. $\widetilde{X}_2^*$ is input into the convolutional layer with the filter size $7 \times 7$ followed by a batch normalization layer. Then, we also apply the sigmoid function on the output $\widetilde{X}_2^{**}$ of the last step to obtain the attention weight, which is used for rescaling the feature maps $\widetilde{X}_2$. The dimension order of the rescaled feature maps is exchanged to retain the same shape as input $X_2$. The final output of second module is $\widetilde{X}_2^{***}$.

3.3.2. Spatial Attention Block (SAB)

The previous literature of SR only focuses on the inter-channel relationship of the features. As discussed in [25], we use the spatial attention module (see Figure 3) to complement with CDIB for exploiting the inter-spatial correlations of the features. The spatial attention tells the network which pivotal part should be focused or suppressed. The spatial attention maps are further generated by the following operations. First, we apply the max-pooling and average-pooling operations on the feature maps $X_3 \in \mathbb{R}^{C \times W \times H}$ along the zero-th dimension and concatenate the outputs to generate a useful feature map. The combined output is convolved with the convolutional layer with the filter size $7 \times 7$.

The above output is also passed through a batch normalization layer. In brief, the spatial attention weight is expressed as follows.

$$X_3^* = SIGMOID\left(H_{BN}\left(f^{7\times7}\left(H_{CAT}\left(H_{MP_{0d}}\left(\widetilde{X}_1\right), H_{AP_{0d}}\left(\widetilde{X}_1\right)\right)\right)\right)\right), \tag{15}$$

where $SIGMOID$ denotes the sigmoid function, $H_{BN}(\cdot)$ denotes the batch normalization, $f^{7\times7}$ represents the convolutional layer with the filter size $7 \times 7$, $H_{CAT}$ represents the function which concatenates the given input sequences in the specified dimension, $H_{MP_{0d}}$ and $H_{AP_{0d}}$ represent the max-pooling and average-pooling function in the zero-th dimension, respectively.

The spatial attention weight is applied to the input for obtaining the final output.

$$X_3^{**} = X_3^* \cdot X_3, \tag{16}$$

### 3.3.3. Feature Aggregation Method

The refined outputs of $CDIB_1$, $CDIB_2$, and SAB are further aggregated by assigning the appropriate scale factor. Then, we get the output of the triplet attention module.

$$Y = \alpha\widetilde{X}_1^{***} + \beta\widetilde{X}_2^{***} + \gamma X_3^{**}, \tag{17}$$

Please note that we adopt the simple averaging aggregation, where $\alpha = \beta = \gamma = \frac{1}{3}$.

### 3.4. Residual Triplet Attention Module (RTAM)

As discussed in Section 3.2, the residual learning and shortcut connections ease the difficulty of learning between the LR and HR image. Similarly, inspired by the successful application of the residual blocks in [22], we use this module in SR. As shown in Figure 3, we embed the TAM in the basic residual module and propose the RTAM. The mathematical expression for the $n$-th RTAM in $m$-th NRG is expressed as

$$M_{m,n} = F_{TAM,m,n}(X_{m,n}) + M_{m,n-1}, \tag{18}$$

where $M_{m,n}$ and $M_{m,n-1}$ denote the output and input of the RTAM. $F_{TAM,m,n}$ denotes the corresponding function of the triplet attention. $X_{m,n}$ denotes the residual part which is formulated as

$$X_{m,n} = W_2(RELU(W_1(M_{m,n-1}))), \tag{19}$$

where $W_1$ and $W_2$ denote the weights of the two convolutional layers in $RTAM$, respectively. $RELU$ denotes the $RELU$ function.

## 4. Discussion
### 4.1. Difference between RTAN and RCAN

Zhang et al. [6] proposed the RCAN by introducing the channel attention mechanism, which made a significant improvement in SR performance. The main differences between the RCAN and the proposed RTAN are listed as follows. First, although both RCAN and RTAN adopt residual learning, RCAN builds a very deep network (more than 400 layers), while the network depth of our proposed RTAN is much shallower than that of RCAN (about 240 layers). Second, the most crucial module of the RCAN is the stacked residual channel attention blocks, RCAN only considers the interdependencies among feature channels. While the key part of our RTAN is the residual triplet attention module (RTAM). The RTAM can explicitly model the cross-dimensional feature interdependencies and interactions. Finally, the advantage of RCAN is that it introduces an effective channel attention mechanism and constructs a very deep network to improve the quality of image reconstruction. However, RCAN also has some disadvantages. It is too complex to be used in practice and it only explores feature information in the channel dimension.

## 4.2. Advantages of the RTAN over RCAN

The advantages of our RTAN over RCAN lie in the following aspects. First, for the real-world image super-resolution, the number of parameters of RTAN is 9.6M, far smaller than 16M in RCAN, which is more conducive to the practical applications. Moreover, the parameter of the channel attention module is about 624 K, while the parameter of the residual triplet attention module is only about 354 K. Second, for RCAN, the operation of global average pooling in the channel attention module leads to the loss of spatial information. While our RTAM uses both the average-pooling and the max-pooling operations to preserve the spatial information. Third, the RTAM utilizes the inherent information of the features by blending different dimensional information with less parameters. In addition, the RTAM introduces the spatial attention auxiliary branch to further enhance the spatial discrimination ability of the network.

## 5. Experiments and Analysis

### 5.1. Datasets and Evaluation Metrics

#### 5.1.1. Datasets

In this paper, we use the DIV2K [64] dataset and the real-world [65] SR dataset for training the proposed network. The DIV2K dataset comprises 1000 images for training, validation, and testing. The real-world SR dataset comprises 595 pairs of HR-LR images which are collected by two DSLR cameras. We use the real-world SR dataset version 1. For testing the network performance, we use five commonly used benchmark datasets, namely: Set5 [66], Set14 [67], BSD100 [68], Urban100 [13], and Manga109 [69]. Moreover, we also use the test images which have 30 HR-LR image pairs from the real-world SR dataset. The main characteristics of the datasets are present in Table 1. The bicubic (BI) [7], blur-downscale (BD) [7], and the real-world degradation models [65] are adopted to perform the experiments.

**Table 1.** The main characteristics of the public datasets.

| Datasets | Amount | Format | Key Category |
|---|---|---|---|
| DIV2K | 1000 | PNG | animals, plants, architecture, scenery, outdoor environment, etc. |
| Set5 | 5 | PNG | baby, bird, butterfly, head, woman |
| Set14 | 14 | PNG | animals, people, flowers, vegetables, boats, bridges, slides, etc. |
| BSD100 | 100 | PNG | animals, plants, landscapes, buildings, etc. |
| Urban100 | 100 | PNG | buildings, city, etc. |
| Manga109 | 109 | PNG | manga volume |
| Real-world SR | 178 | PNG | architecture, plants, environment, handmade items, etc. |

#### 5.1.2. Evaluation Metrics

The reconstructed images are transformed to YCbCr space. Then, we evaluate the results with two quantitative metrics, i.e., peak signal-to-noise ratio (PSNR) and structural similarity index (SSIM) [70] on Y channel.

### 5.2. Implementation and Training Details

During the training process, we perform data augmentation on all the training images. This is accomplished by rotating the images by $90°$, $180°$, $270°$ randomly and by flipping them horizontally. The LR and HR images are cropped into appropriate patches with the size of $48 \times 48$ to enlarge the training datasets. We further set the mini-batch size as 16. The proposed RTAN is trained by adopting the ADAM [71] optimizer with $\beta_1 = 0.9$, $\beta_2 = 0.999$, and $\varepsilon = 10^{-8}$. We initialize the learning rate as $1 \times 10^{-4}$ and then the learning rate decreases half every 200 epochs. For training the BI and BD degradation model, we set NRG number as M = 10. Within each NRG, we set RTAM number as N = 15. While

training the real-world degradation model, we set NRG number as M = 10 and RTAM number N = 12. We set the maximum number of steps to $1 \times 10^6$. It takes about 11 days and 10 days to train RTAN on DIV2K and real-world SR datasets, respectively. The models are trained using the PyTorch [72] framework. All the experiments are conducted on an Nvidia 2080Ti GPU.

*5.3. Ablation Study*

To demonstrate the effect of the RTAN, we perform a series of ablation experiments to compare the effectiveness of different modules, including GSC, LSC, CDIB, and SAB. The performance on Set5 × 4 is shown in Table 1.

### 5.3.1. GSC and LSC

To prove the effect of the MNRG structure, we remove the GSC or/and LSC from the proposed network. $R_{base}$ is the base model which only contains over 300 convolutional layers with 10 NRGs and 15 RTAMs in each NRG. As presented in Table 2, the PSNR in $R_{base}$ is extremely low. This indicates that simply stacking the single-layer residual structure is not sufficient to improve the SR performance effectively. In comparison, the PSNR from $R_a$ to $R_c$ has a sustaining boost without introducing any extra parameters. Specifically, $R_a$ and $R_b$ obtain 0.31 dB and 0.39 dB PSNR gain over $R_{base}$, respectively. When both of GSC and LSC are added to $R_{base}$, the PSNR increases by 0.41 dB. The results show that the MNRG structure results into a huge improvement in the SR performance and makes the network training easier. This is because the GSC and LSC bypass the redundant low-frequency information and reuse the lower layer information for very deep networks.

**Table 2.** The ablation results of the key components (i.e., CDIB$_1$, CDIB$_2$, SAB, GSC, and LSC). The best PSNR (dB) is tested on Set5 (×4) in 100 epochs.

| | $R_{base}$ | $R_a$ | $R_b$ | $R_c$ | $R_d$ | $R_e$ | $R_f$ | $R_g$ | $R_h$ |
|---|---|---|---|---|---|---|---|---|---|
| CDIB$_1$ | × | × | × | × | × | × | √ | √ | √ |
| CDIB$_2$ | × | × | × | × | × | √ | × | √ | √ |
| SAB | × | × | × | × | √ | × | × | × | √ |
| GSC | × | √ | × | √ | √ | √ | √ | √ | √ |
| LSC | × | × | √ | √ | √ | √ | √ | √ | √ |
| Params(K) | 11,809 | 11,809 | 11,809 | 11,809 | 11,854 | 11,839 | 11,839 | 11,839 | 11,854 |
| PSNR | 31.79 | 32.10 | 32.18 | 32.20 | 32.21 | 32.21 | 32.23 | 32.26 | 32.27 |

### 5.3.2. SAB and CDIB

We further evaluate the effect of the spatial attention block and cross-dimensional interaction block based on the above ablation investigations. The simulation results from $R_d$ to $R_f$ demonstrate the effectiveness of the individual module. We observe that $R_d$ performs better than $R_c$. Similarly, SAB slightly improves the performance from 32.20 dB to 32.21 dB. Please note that both $R_e$ and $R_f$ achieve 0.01 dB and 0.03 dB PSNR gains, respectively, in comparison with $R_c$ by the introduction of 30 K parameters only. It is worth noting that by using both CDIB$_1$ and CDIB$_2$, the performance of $R_g$ improves significantly as compared to the methods of $R_a$ to $R_f$. Besides, when all the components are added to the base model ($R_{base}$), the proposed RTAN ($R_h$) results in a huge improvement of 0.48 dB in PSNR as compared to $R_{base}$. These comparisons firmly indicate that the cross-dimensional interaction and the inter-spatial correlation of the features play a significant role in improving the ability of image reconstruction.

The aforementioned ablation investigations demonstrate the rationality and necessity of the components of the proposed network. The proposed RTAN model shows the superiority in the SR performance.

*5.4. Comparison with State-of-the-Art*

5.4.1. Results with Bicubic (BI) Degradation Model

The BI degradation model has been widely used to obtain LR images in the image SR tasks. In order to demonstrate the effectiveness of the RTAN, we compared it with 16 state-of-the-art CNN-based SR methods, including SRMDNF [7], NLRN [32], EDSR [17], DBPN [73], NDRCN [35], ACNet [38], FALSR-A [37], OISR-RK2-s [34], MCAN [47], A$^2$F-SD [48], A2N-M [63], DeFiAN$_S$ [61], IMDN [33], SMSR [36], PAN [59], MGAN [62], RNAN [55].

Table 3 shows all the quantitative results for ×2, ×3, ×4 scaling factors. In general, as compared with all the methods presented in the literature, the proposed model shows the best performance on most of the standard benchmark datasets for various scaling factors. As the scaling factor increases from 2 to 4, the proposed RTAN performs significantly better than other methods. Particularly when compared with MGAN, RTAN exceeds MGAN by a margin of 0.31 dB PSNR on Urban100 × 4, while the number of parameters of the RTAN is similar to that of MGAN. In addition, the PSNR and SSIM are higher than EDSR on most benchmark datasets, while the number of parameters of RTAN (11.6M) is far smaller than that of EDSR (43M). This competitive performance indicates that RTAN makes better use of the limited feature information by employing the efficient network structure. This is mainly because the RTAN adopts the spatial attention and cross-dimensional interaction mechanism to enable the network to exploit the cross-dimensional interactive features and spatial information.

**Table 3.** The quantitative comparison (i.e., average PSNR/SSIM) with BI degradation model on datasets Set5, Set14, BSD100, Urban100, and Manga109. The best and second-best results are highlighted and underlined, respectively.

| Methods | Scale | Set5 | Set14 | BSD100 | Urban100 | Manga109 |
|---|---|---|---|---|---|---|
| | | PSNR/SSIM | PSNR/SSIM | PSNR/SSIM | PSNR/SSIM | PSNR/SSIM |
| Bicubic | 2 | 33.66/0.9299 | 30.24/0.8688 | 29.56/0.8431 | 26.88/0.8403 | 30.80/0.9339 |
| SRMDNF | 2 | 37.79/0.9601 | 33.32/0.9159 | 32.05/0.8985 | 31.33/0.9204 | 38.07/0.9761 |
| NLRN | 2 | 38.00/0.9603 | 33.46/0.9159 | 32.19/0.8992 | 31.81/0.9246 | −/− |
| DBPN | 2 | 38.09/0.9600 | 33.85/0.9190 | 32.27/0.9000 | 32.55/0.9324 | 38.89/0.9775 |
| EDSR | 2 | 38.11/0.9601 | 33.92/0.9195 | 32.32/0.9013 | 32.93/0.9351 | 39.10/0.9773 |
| NDRCN | 2 | 37.73/0.9596 | 33.20/0.9141 | 32.00/0.8975 | 31.06/0.9175 | −/− |
| FALSR-A | 2 | 37.82/0.9595 | 33.55/0.9168 | 32.12/0.8987 | 31.93/0.9256 | −/− |
| OISR-RK2-s | 2 | 37.98/0.9604 | 33.58/0.9172 | 32.18/0.8996 | 32.09/0.9281 | −/− |
| MCAN | 2 | 37.91/0.9597 | 33.69/0.9183 | 32.18/0.8994 | 32.46/0.9303 | −/− |
| A$^2$F-SD | 2 | 37.91/0.9602 | 33.45/0.9164 | 32.08/0.8986 | 31.79/0.9246 | 38.52/0.9767 |
| DeFiAN$_S$ | 2 | 38.03/0.9605 | 33.63/0.9181 | 32.20/0.8999 | 32.20/0.9286 | 38.91/0.9775 |
| A2N-M | 2 | 38.06/0.9601 | 33.73/0.9190 | 32.22/0.8997 | 32.34/0.9300 | 38.80/0.9765 |
| IMDN | 2 | 38.00/0.9605 | 33.63/0.9177 | 32.19/0.8996 | 32.17/0.9283 | 38.88/0.9774 |
| SMSR | 2 | 38.00/0.9601 | 33.64/0.9179 | 32.17/0.8990 | 32.19/0.9284 | 38.76/0.9771 |
| PAN | 2 | 38.00/0.9605 | 33.59/0.9181 | 32.18/0.8997 | 32.01/0.9273 | 38.70/0.9773 |
| MGAN | 2 | 38.16/**0.9612** | 33.83/0.9198 | 32.28/0.9009 | 32.75/0.9340 | 39.11/0.9778 |
| RNAN | 2 | 38.17/0.9611 | 33.87/0.9207 | 32.32/0.9014 | 32.73/0.9340 | 39.23/**0.9785** |
| RTAN(Ours) | 2 | **38.19**/0.9605 | **34.07**/**0.9211** | **32.37**/**0.9016** | **33.18**/**0.9372** | **39.34**/0.9779 |
| Bicubic | 3 | 30.39/0.8682 | 27.55/0.7742 | 27.21/0.7385 | 24.46/0.7349 | 26.95/0.8556 |
| SRMDNF | 3 | 34.12/0.9254 | 30.04/0.8382 | 28.97/0.8025 | 27.57/0.8398 | 33.00/0.9403 |
| NLRN | 3 | 34.27/0.9266 | 30.16/0.8374 | 29.06/0.8026 | 27.93/0.8453 | −/− |
| EDSR | 3 | 34.65/0.9282 | 30.52/0.8462 | 29.25/**0.8093** | 28.80/0.8653 | **34.17**/**0.9476** |
| NDRCN | 3 | 33.90/0.9235 | 29.88/0.8333 | 28.86/0.7991 | 27.23/0.8312 | −/− |
| OISR-RK2-s | 3 | 34.43/0.9273 | 30.33/0.8420 | 29.10/0.8053 | 28.20/0.8534 | −/− |

**Table 3.** *Cont.*

| Methods | Scale | Set5 | Set14 | BSD100 | Urban100 | Manga109 |
|---|---|---|---|---|---|---|
| | | PSNR/SSIM | PSNR/SSIM | PSNR/SSIM | PSNR/SSIM | PSNR/SSIM |
| DeFiAN$_S$ | 3 | 34.42/0.9273 | 30.34/0.8410 | 29.12/0.8053 | 28.20/0.8528 | 33.72/0.9447 |
| A$^2$F-SD | 3 | 34.23/0.9259 | 30.22/0.8395 | 29.01/0.8028 | 27.91/0.8465 | 33.29/0.9424 |
| A2N-M | 3 | 34.50/0.9279 | 30.41/0.8438 | 29.13/0.8058 | 28.35/0.8563 | 33.79/0.9458 |
| IMDN | 3 | 34.36/0.9270 | 30.32/0.8417 | 29.09/0.8046 | 28.17/0.8519 | 33.61/0.9445 |
| SMSR | 3 | 34.40/0.9270 | 30.33/0.8412 | 29.10/0.8050 | 28.25/0.8536 | 33.68/0.9445 |
| PAN | 3 | 34.40/0.9271 | 30.36/0.8423 | 29.11/0.8050 | 28.11/0.8511 | 33.61/0.9448 |
| MCAN | 3 | 34.45/0.9271 | 30.43/0.8433 | 29.14/0.8060 | 28.47/0.8580 | −/− |
| MGAN | 3 | <u>34.65</u>/**0.9292** | 30.51/0.8460 | 29.22/<u>0.8086</u> | 28.61/0.8621 | 34.00/<u>0.9474</u> |
| RTAN(Ours) | 3 | **34.75**/<u>0.9288</u> | **30.60**/**0.8468** | **29.28**/**0.8093** | **28.88**/**0.8669** | <u>34.05</u>/<u>0.9474</u> |
| Bicubic | 4 | 28.42/0.8104 | 26.00/0.7027 | 25.96/0.6675 | 23.14/0.6577 | 24.89/0.7866 |
| SRMDNF | 4 | 31.96/0.8925 | 28.35/0.7787 | 27.49/0.7337 | 25.68/0.7731 | 30.09/0.9024 |
| NLRN | 4 | 31.92/0.8916 | 28.36/0.7745 | 27.48/0.7346 | 25.79/0.7729 | −/− |
| DBPN | 4 | 32.47/0.8980 | 28.82/0.7860 | 27.72/0.7400 | 26.38/0.7946 | 30.91/0.9137 |
| EDSR | 4 | 32.46/0.8968 | 28.80/<u>0.7876</u> | 27.71/0.7420 | <u>26.64</u>/<u>0.8033</u> | 31.02/0.9148 |
| NDRCN | 4 | 31.50/0.8859 | 28.10/0.7697 | 27.30/0.7263 | 25.16/0.7546 | −/− |
| OISR-RK2-s | 4 | 32.21/0.8950 | 28.63/0.7822 | 27.58/0.7364 | 26.14/0.7874 | −/− |
| DeFiAN$_S$ | 4 | 32.16/0.8942 | 28.63/0.7810 | 27.58/0.7363 | 26.10/0.7862 | 30.59/0.9084 |
| A$^2$F-SD | 4 | 32.06/0.8928 | 28.47/0.7790 | 27.48/0.7373 | 25.80/0.7767 | 30.16/0.9038 |
| A2N-M | 4 | 32.27/0.8963 | 28.69/0.7842 | 27.61/0.7376 | 26.28/0.7919 | 30.59/0.9103 |
| IMDN | 4 | 32.21/0.8948 | 28.58/0.7811 | 27.56/0.7353 | 26.04/0.7838 | 30.45/0.9075 |
| SMSR | 4 | 32.12/0.8932 | 28.55/0.7808 | 27.55/0.7351 | 26.11/0.7868 | 30.54/0.9085 |
| PAN | 4 | 32.13/0.8948 | 28.61/0.7822 | 27.59/0.7363 | 26.11/0.7854 | 30.51/0.9095 |
| MCAN | 4 | 32.33/0.8959 | 28.72/0.7835 | 27.63/0.7378 | 26.43/0.7953 | −/− |
| MGAN | 4 | 32.45/0.8980 | 28.74/0.7852 | 27.68/0.7400 | 26.47/0.7981 | 30.81/0.9131 |
| RNAN | 4 | <u>32.49</u>/<u>0.8982</u> | <u>28.83</u>/**0.7878** | <u>27.72</u>/<u>0.7421</u> | 26.61/0.8023 | <u>31.09</u>/<u>0.9149</u> |
| RTAN(Ours) | 4 | **32.61**/**0.8987** | **28.84**/0.7873 | **27.77**/**0.7422** | **26.78**/**0.8068** | **31.11**/**0.9156** |

Figure 4 shows the visual comparisons of several recent CNN-based SISR methods. The SR results of the "img_004" from Urban100 are presented. It is evident that the outputs of most of the compared methods are prone to the heavy blurring of artifacts. Additionally, most of these methods are unable to recover the detailed structure of metal holes and green lines. In contrast, the proposed RTAN produces the clearest image, which is more similar to the ground truth. Similar phenomena are evident in image "img_074", where other SR methods do not replicate the textures of the grids and cannot successfully resolve the aliasing problems. Additionally, it is noteworthy that most of these methods distort the original structures and generate blurred artifacts. However, as compared with other methods, the RTAN alleviates blurred artifacts and achieves sharper results. These results firmly prove the superiority of the RTAN, which not only shows the powerful reconstruction ability, but also achieves better visual SR results with finer structures.

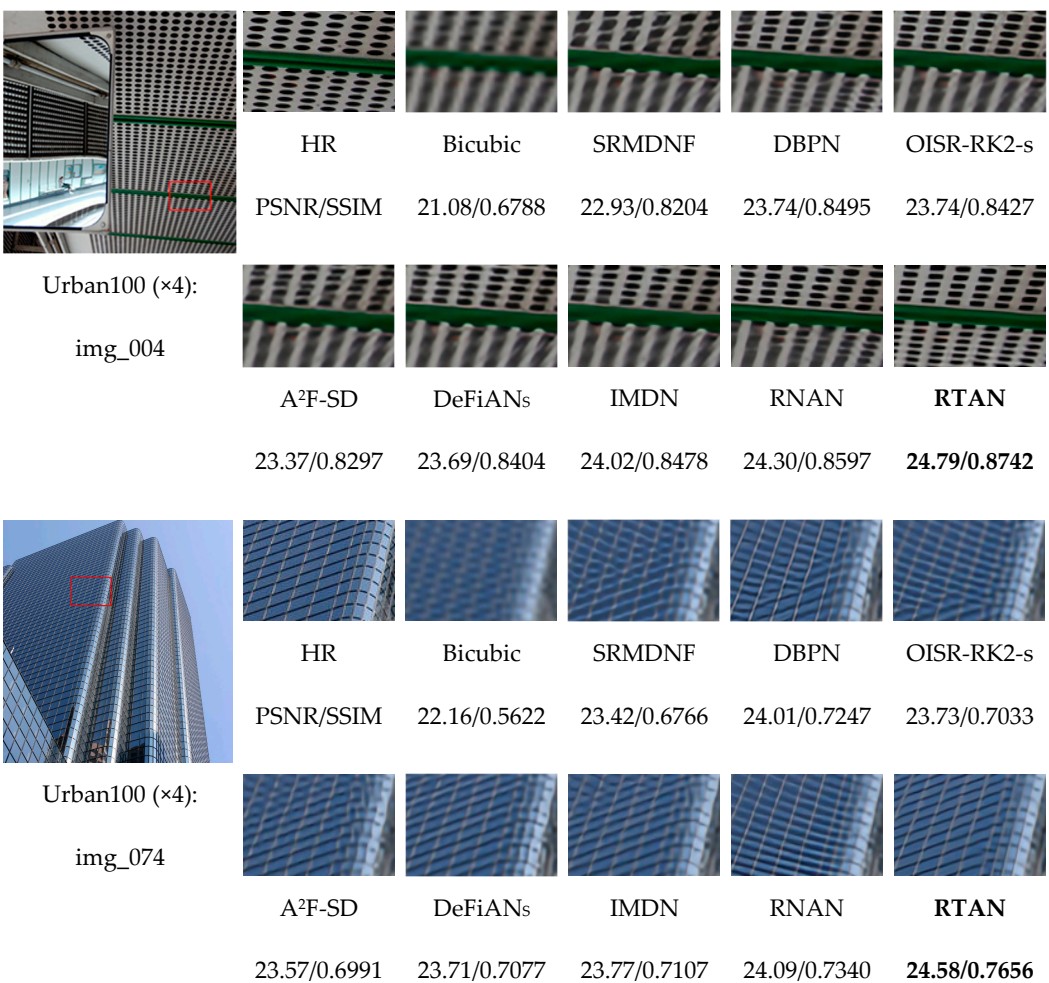

**Figure 4.** The visual comparison of ×4 SR with BI model on Urban100 dataset. The best results are highlighted.

### 5.4.2. Results with Blur-Downscale (BD) Degradation Model

In order to demonstrate the powerful reconstruction ability of the proposed method with BD degradation model, we compare the RTAN with 14 state-of-the-art CNN-based models, i.e., SPMSR [4], SRCNN [5], FSRCNN [74], VDSR [12], SRMD [7], EDSR [17], RDN [16], IRCNN [75], SRFBN [76], RCAN [6], A$^2$F-SD [48], IMDN [33], DeFiAN$_S$ [61], PAN [59], and MGAN [62].

Considering the work presented in [6,7], we compare the ×3 SR results with blur-downscale (BD) degradation model. As shown in Table 4, in general, the proposed RTAN obtains similar results to RCAN and outperforms other state-of-the-art methods. As compared with RCAN, the proposed RTAN performs better in terms of PSNR for all datasets, while the performance in terms of SSIM for the datasets, e.g., Set5, BSD100, Manga109 is similar. Specially, the proposed RTAN achieves 0.09 dB PSNR gain over RCAN on Urban100 dataset. The simulation results show that the proposed RTAN has a greater ability to deal with more complex degradation models.

We also provide the visual comparisons on ×3 scaling factor with BD degradation model in Figure 5. For reconstructing the detailed textures of zebras in image "253027", most of the other methods show aliasing effects and significantly blurred artifacts. In contrast, the proposed RTAN recovers information closer to the ground truth. Particularly, the RCAN generates intersecting zebra stripes, while the proposed method restores the texture consistent with the ground truth. These results firmly demonstrate the superiority of the proposed method in alleviating the blurring artifacts.

**Table 4.** The quantitative comparison (i.e., average PSNR/SSIM) with BD degradation model on datasets Set5, Set14, BSD100, Urban100, and Manga109. The best and second-best results are highlighted and <u>underlined</u>, respectively.

| Methods | Scale | Set5 | Set14 | BSD100 | Urban100 | Manga109 |
|---|---|---|---|---|---|---|
| | | PSNR/SSIM | PSNR/SSIM | PSNR/SSIM | PSNR/SSIM | PSNR/SSIM |
| Bicubic | 3 | 28.78/0.8308 | 26.38/0.7271 | 26.33/0.6918 | 23.52/0.6862 | 25.46/0.8149 |
| SPMSR | 3 | 32.21/0.9001 | 28.89/0.8105 | 28.13/0.7740 | 25.84/0.7856 | 29.64/0.9003 |
| SRCNN | 3 | 32.05/0.8944 | 28.80/0.8074 | 28.13/0.7736 | 25.70/0.7770 | 29.47/0.8924 |
| FSRCNN | 3 | 26.23/0.8124 | 24.44/0.7106 | 24.86/0.6832 | 22.04/0.6745 | 23.04/0.7927 |
| VDSR | 3 | 33.25/0.9150 | 29.46/0.8244 | 28.57/0.7893 | 26.61/0.8136 | 31.06/0.9234 |
| EDSR | 3 | 34.69/0.9278 | 30.58/0.8447 | 29.27/0.8083 | 28.64/0.8611 | 34.24/0.9470 |
| IRCNN | 3 | 33.38/0.9182 | 29.63/0.8281 | 28.65/0.7922 | 26.77/0.8154 | 31.15/0.9245 |
| SRMD | 3 | 34.01/0.9242 | 30.11/0.8364 | 28.98/0.8009 | 27.50/0.8370 | 32.97/0.9391 |
| RDN | 3 | 34.58/0.9280 | 30.53/0.8447 | 29.23/0.8079 | 28.46/0.8582 | 33.97/0.9465 |
| SRFBN | 3 | 34.66/0.9283 | 30.48/0.8439 | 29.21/0.8069 | 28.48/0.8581 | 34.07/0.9466 |
| A$^2$F-SD | 3 | 33.81/0.9217 | 29.96/0.8337 | 28.84/0.7975 | 27.20/0.8285 | 32.54/0.9351 |
| IMDN | 3 | 34.35/0.9254 | 30.31/0.8392 | 29.08/0.8029 | 28.03/0.8472 | 33.66/0.9427 |
| DeFiAN$_S$ | 3 | 34.34/0.9252 | 30.32/0.8396 | 29.08/0.8030 | 28.06/0.8478 | 33.68/0.9426 |
| PAN | 3 | 34.32/0.9260 | 30.33/0.8406 | 29.08/0.8037 | 27.93/0.8462 | 33.46/0.9431 |
| RCAN | 3 | <u>34.70</u>/**0.9288** | <u>30.63</u>/**0.8462** | <u>29.32</u>/**0.8093** | <u>28.81</u>/<u>0.8647</u> | 34.38/**0.9483** |
| MGAN | 3 | 34.63/0.9284 | 30.54/0.8450 | 29.24/0.8081 | 28.51/0.8580 | 34.11/0.9467 |
| RTAN(Ours) | 3 | **34.76**/<u>0.9285</u> | **30.66**/**0.8462** | **29.33**/<u>0.8090</u> | **28.90**/**0.8649** | **34.39**/<u>0.9478</u> |

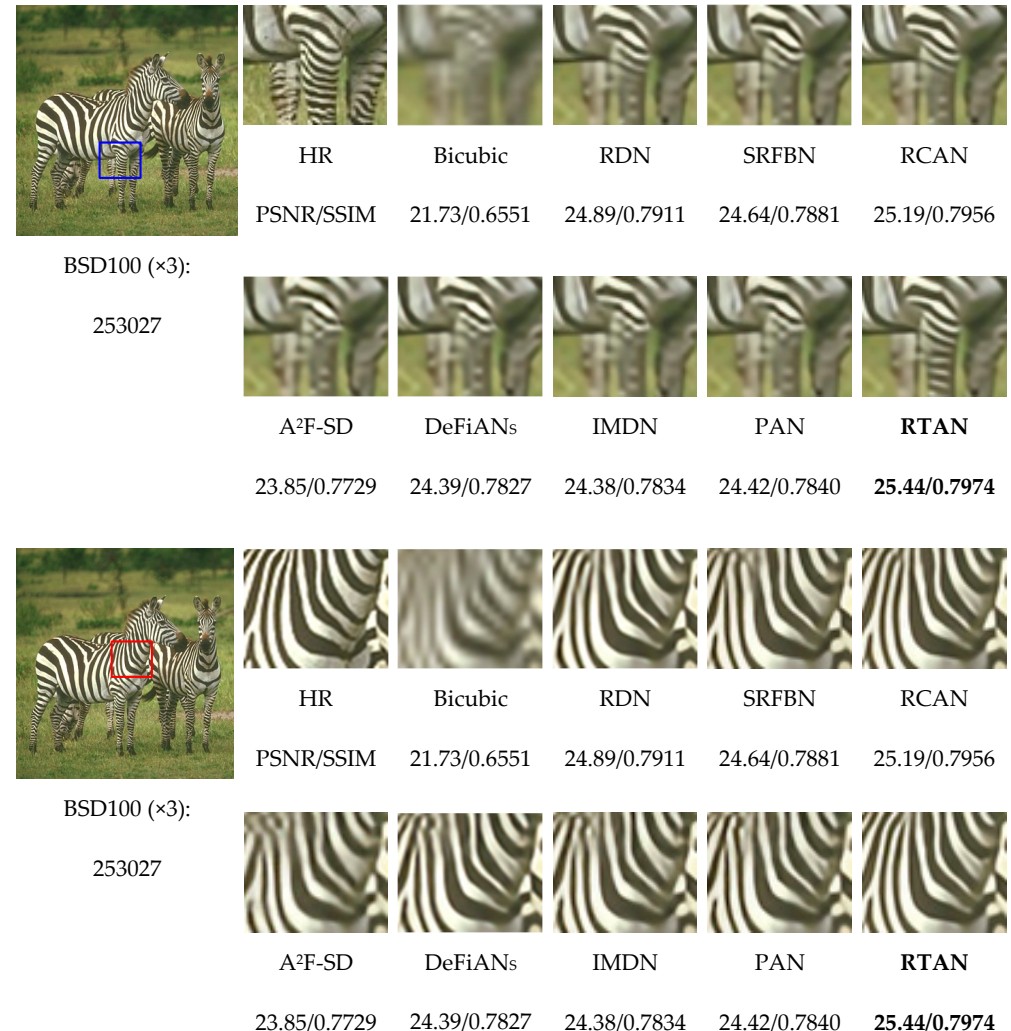

**Figure 5.** The visual comparisons of ×3 SR with BD degradation model on BSD100 dataset. The best results are highlighted.

### 5.4.3. Results with the Real-World Degradation Model

To further evaluate the performance of the RTAN, we provide the $\times 4$ SR results with the challenging real-world degradation model. Contrary to the simulated degradation model, i.e., BI and BD degradation, which usually deviate from the complex real world, the real-world degradation model is more applicable to the practical applications. We compare the proposed RTAN with eight representative SR models of different sizes, including CARN [31], RCAN [6], EDSR [17], RNAN [55], LP-KPN [65], DeFiAN$_S$ [61], and MIRNet [52].

As shown in Table 5, we observe that the RTAN achieves the best performance on the two real-world test datasets. It is evident that the proposed RTAN obtains notable performance gains and has the highest PSRN/SSIM as compared to other methods. Specifically, the PSNR gain of RTAN in comparison with RCAN and EDSR are up to 0.22 dB and 0.51 dB on Nikon dataset, respectively. As compared with the RCAN which introduces the channel attention, the proposed RTAN achieves better results in terms of PSNR and SSIM. Moreover, as compared with the RNAN and MIRNet, our RTAN exceeds them by a margin of 0.25 and 0.23 PSNR on Nikon dataset, respectively. Furthermore, the proposed method outperforms CARN by a large margin of 0.79 dB PSNR on Canon dataset. These comparisons indicate that the proposed method has the ability to generalize for practical applications under the complex real-world degradation.

**Table 5.** The quantitative results (i.e., average PSNR/SSIM) with real-world degradation model on datasets Canon and Nikon. The best results are highlighted.

| Methods | Params | Scale | Canon PSNR/SSIM | Nikon PSNR/SSIM |
|---------|--------|-------|-----------------|-----------------|
| CARN | 1.59M | 4 | 29.14/0.8336 | 28.42/0.7931 |
| EDSR | 43M | 4 | 29.56/0.8249 | 28.35/0.7986 |
| RCAN | 16M | 4 | 29.79/0.8473 | 28.64/0.8035 |
| IMDN | 0.72M | 4 | 28.95/0.8371 | 28.50/0.7916 |
| RNAN | 9.3M | 4 | 29.44/0.8408 | 28.61/0.8029 |
| LP-KPN | 5.73M | 4 | 29.52/0.8430 | 27.54/0.7923 |
| DeFiANs | 1.06M | 4 | 29.37/0.8390 | 28.60/0.7967 |
| MIRNet | 31.8M | 4 | 29.86/0.8495 | 28.63/0.8034 |
| RTAN(Ours) | 9.6M | 4 | **29.93/0.8504** | **28.86/0.8071** |

Figure 6 shows the visual SR results of RTAN and other competing methods for the real-world task. For recovering the challenging and tiny texture in image "Canon 006", it is noteworthy that other methods recover some details of the contours to some degree; however, they still suffer from the serious blurring of artifacts. Contrarily, the proposed RTAN yields more convincing details which are clearer and comprise small number of artifacts than other models. Similar observations are further shown in images "Nikon 014" and "Nikon 015", where the compared methods fail to recover the minute details from the target patterns. In contrast, the proposed method splits the gap of lines in a clear manner. This indicates that the proposed RTAN has the ability to produce more accurate information and reconstructs the texture which is faithful to the ground truth. Overall, the consistent efficient SR results demonstrate the superiority of the RTAN which has a strong reconstruction ability for the real-world degradation model.

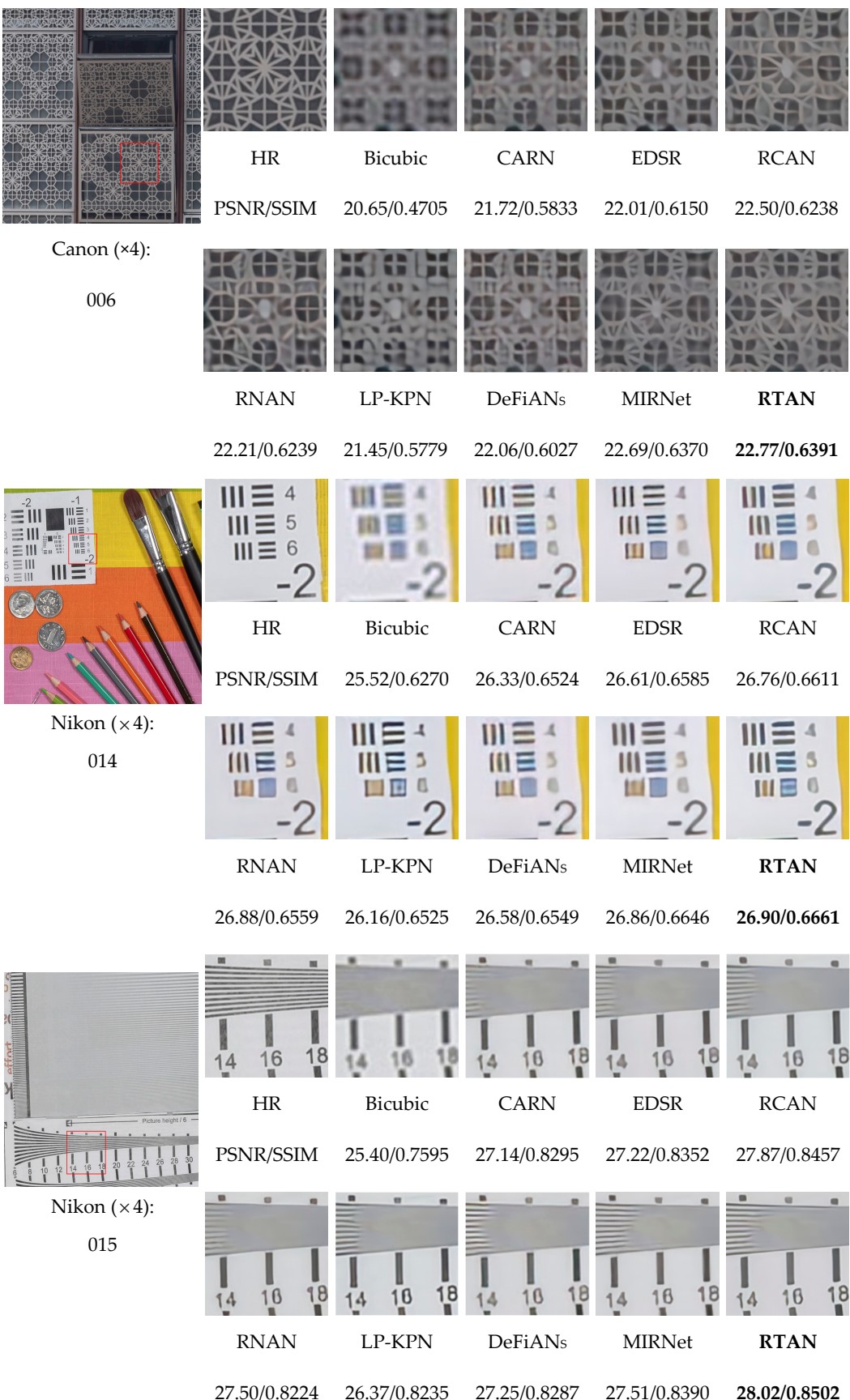

**Figure 6.** The visual comparisons for ×4 SR with real-world degradation model on Canon and Nikon datasets. The best results are highlighted.

### 5.5. Model Complexity Comparison

We also compare the proposed RTAN with other representative models in terms of the SR performance, model size, and computational cost. Considering [6,31], we measure the size of the models with the number of parameters. FLOPs [77] and Mult-Adds are two methods to measure the computational efficiency of the model. Following most works [31,78], for evaluating the models in terms of computational cost, we use the number of composite multiply accumulate operations (Mult-Adds) by assuming the resolution of reconstructed image to be 720p (1280 × 720).

#### 5.5.1. Model Size Analysis

Figure 7a,b show the size of models and SR performance of recent state-of-the-art deep CNN-based SR algorithms under the BI and real-world degradation models, respectively. In order to make comprehensive evaluation, the SR image quality metric is calculated by the average of PSNR on four standard benchmark datasets, i.e., Set5, Set14, B100, Urban100 and two realistic test datasets, i.e., Canon, Nikon, respectively. In Figure 7a, it is obvious that the RTAN outperforms all state-of-the-art models. Especially, the EDSR model which has around 43M parameters has nearly four times more parameters than the proposed RTAN (11.6M). While the PSNR of EDSR is much lower than the proposed RTAN on four benchmark datasets. Similarly, the number of parameters of RTAN is less than RDN (22.1M), but obtains better performance. In addition, the RTAN has similar number of parameters to MGAN (11.7M) but the former achieves much higher PSNR. Considering the importance of practical applications, we further reduce the size of the proposed model for the real-world scenes. As shown in Figure 7b, it not surprising that the CARN and IMDN contain significantly small number of parameters. However, this comes with performance degradation. However, the proposed RTAN has fewer parameters (9.6M) than that of RCAN (16M), MIRNet (31.8M), and EDSR (43M) but achieves much higher PSNR. This implies that the RTAN maintains a better trade-off between model size and performance.

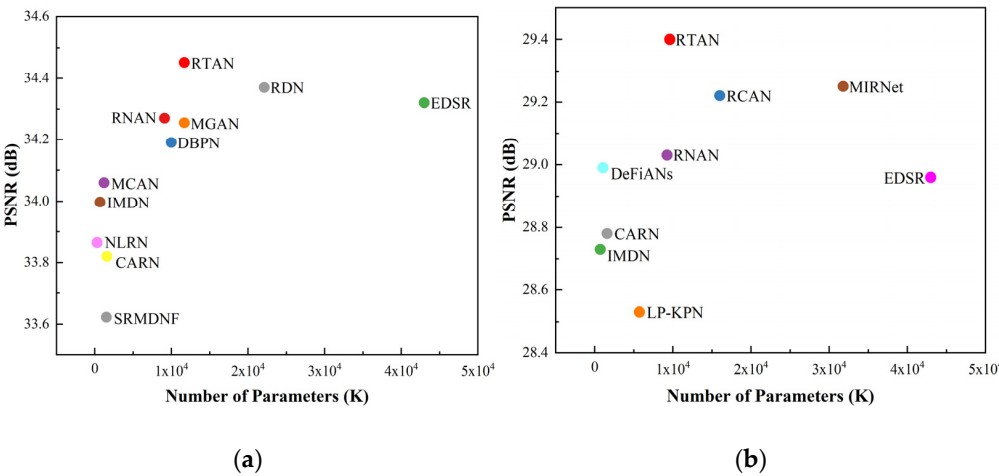

(**a**)                                          (**b**)

**Figure 7.** The performance and number of parameters. The results are evaluated on four standard benchmark datasets (Set5, Set14, BSD100, Urban100) and two real-world test datasets (Canon, Nikon). (**a**) Results on BI degradation model (×2), (**b**) Results on real-world degradation model (×4).

#### 5.5.2. Model Computation Cost Analysis

We further measure the computational efficiency of each method for different SR tasks, as shown in Figure 8a,b. From Figure 8a, we observe that the proposed RTAN achieves best performance with less Mult-Adds as compared to DBPN, DRRN, RDN, EDSR, and DRCN. Similar comparisons are further presented in Figure 8b, which shows that the proposed RTAN is more efficient than RCAN, MIRNet, and EDSR. As compared with other larger models, the proposed RTAN attains the balance between the performance and the computational complexity which is considerable from practical applications point of view.

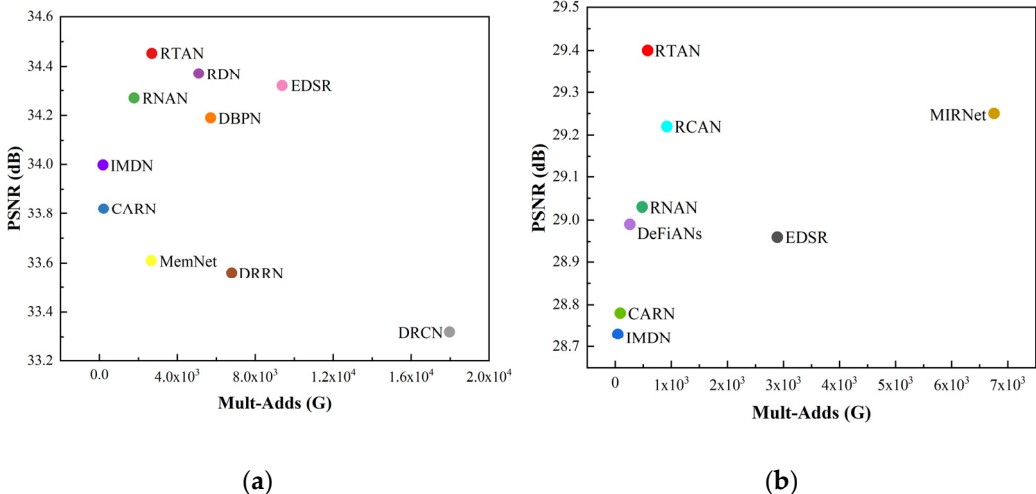

(**a**)                                    (**b**)

**Figure 8.** The performance and number of operations. The results are evaluated on four standard benchmark datasets (Set5, Set14, BSD100, Urban100) and two real-world test datasets (Canon, Nikon). (**a**) Results on BI degradation model (2), (**b**) Results on real-world degradation model (×4).

## 6. Conclusions

In this work, we propose a novel residual triplet attention network (RTAN) for image super-resolution. Specifically, the multiple nested residual group (MNRG) structure allows the proposed RTAN to stabilize the training process. Meanwhile, the MNRG structure focuses on the high-frequency information of the input image and reuses low-layer feature maps. Furthermore, in order to effectively utilize the advantages of CNNs and consider the correlation between different dimensions of features, the residual triplet attention module (RTAM) which captures the interactions and interdependencies between different dimensions of features in the intermediate layers by using small number of parameters is proposed. Comprehensive evaluations and ablation investigations on benchmark datasets with BI, BD, and real-world degradation models demonstrate the effectiveness of the proposed RTAN. In future research work, we plan to design a network which is more efficient and lightweight to deal with the more complex degradation models of the images.

**Author Contributions:** Conceptualization, Y.S.; methodology, F.H. and Z.W.; investigation, Z.W.; writing—original draft preparation, Z.W.; writing—review and editing, Z.W. and Y.S.; validation, J.W., L.C. and Y.S.; formal analysis, J.W., F.H. and Y.S.; supervision, F.H.; project administration, L.C. and Y.S.; funding acquisition, F.H. All authors have read and agreed to the published version of the manuscript.

**Funding:** This research was funded by National Natural Science Foundation of China (62005049).

**Conflicts of Interest:** The authors declare no conflict of interest.

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
