# Peer review of "Residual Triplet Attention Network for Single-Image Super-Resolution"

_electronics, doi:10.3390/electronics10172072_

Round 1

Reviewer 1 Report

The paper presents an interesting idea but should be improved in some parts:

- The state of the art should be merged into a single section and better focused on the latest proposal with respect to the problem addressed;
- Section 4 should be extended by explaining the advantages introduced and the disadvantages introduced by the two techniques;
- The section relating to datasets should be integrated with a table describing the main characteristics;
- In this context the following paper should be mentioned:
Ullah, I., Manzo, M., Shah, M., & Madden, M. (2019). Graph Convolutional Networks: analysis, improvements and results. arXiv preprint arXiv: 1912.09592.

Reviewer 2 Report

The paper examines “single image super-resolution” (SISR) methods to convert, using convolutional neural networks (CNNs), low-resolution (LR) images into suitable high-resolution (HR) images. The authors develop a multiple nested residual group (MNRG) structure and a light-weight residual triplet attention module (RTAM) . They claim superiority of the proposed RTAN over the state-of-the-art SISR networks in terms of both evaluation metrics and visual results. 

While less complex than RCAN, the RTAN proposed model remains very complex, having about 240 layers and 16M parameters.  Is it truly necessary to have such complex models?  With such complexity, it is impossible to discern what is working well or not well.

While I appreciate all the quantitative results listed, for me, the more important issue is visually, and the paper admits it is “evident that the outputs of most of the compared methods are prone to heavy blurring artifacts.”  This is not surprising when one attempts to go from LR to HR.  The results show that RTAN exceeds other methods, but only by very small amounts.

The authors clearly understand the area and do suitable experiments, and have appropriate citations.  However, as I note below, there are numerous flaws in the presentation.

Specific points:

In literature, there are various efficient methods presented to address this issue, such as interpolation-based method .. ->

In the literature, there are various efficient methods presented to address this issue, such as the interpolation-based method ..

The paper uses the term “super-resolution” several times without distinguishing it from “high resolution  (HR)”

proposed the SRCNN .. - define this; do not assume that all readers will figure out this acronym

....inspired by the effect of the residual structure .. -explain what this is

..multiple nested residual group (MNRG) structure to reduce model degradation and reuse more informative LR features.  In MNRG, we adopt the global shortcut connection (GSC) which serves as the first-layer structure to complete the rough learning of LR image.  - this is a good example of a major flaw in the paper; i.e., it introduces numerous topics and terms with little or no explanation; e.g., here we have all of these undefined ideas: multiple nested residual group (MNRG) structure, model degradation, more informative LR features, global shortcut connection, first-layer structure, and rough learning.

..alleviate the training difficulty causing by the network depth .. - and what is this problem?

..learn abstract residual features. - what are these?  Again, this paper often uses general ideas that are actually objectives, assuming that the tools used (e.g., CNNs) will simply solve all issues.

..enables the proposed network acquire ..

..enables the proposed network to acquire ..

..more blending cross-dimensional feature information.  - another vague idea

..explores the interdependencies and interactions .. - and how is this accomplished?

..bypass most of the low-frequency part in the input LR .. - how and why is this done?

..fully exploit the feature information from the intermediate layers of the network.   - how is this done?

..design the multiple nested residual group (MNRG) structure with multi-layer nested residual module.  - this sounds redundant

..inherent information between the spatial dimension and channel dimension of the features in the intermediate layers, - what is this? How done?

..the image super-resolution has made remarkable progress .. - what exactly IS “image super-resolution“?  It is not making progress; research is.

..The SRCNN comprises three-layer convolutional neural network .. ->

..The SRCNN that comprises a three-layer convolutional neural network ..

Lines 101-103 are not helpful; all they do is list a series of acronyms such as: VDSR, DRCN, DRRN and MemNet

..interpolation on LR input images .. - one should explain what this is and why done

..while having relatively .. ->

..while has relatively ..

..sub-pixel layer. This layer is a learnable up-sampling layer and performs the convolution and reshaping operations.  - again, another poor explanation: 1) what does “sub-pixel” mean?, 2) indeed what are “layers” (the paper has not at all given any basic details of CNNs), 3) why up-sample? 4) what is reshaping?

..developed the EDSR which made a significant improvement of SR performance.  - this simply gives the acronym name and no other details

..remove the unnecessary modules in the residual blocks.  - which makes one wonder why there were such modules in the first place?

..utilize the dense connections by using all the hierarchical features of the convolutional layers.  - more generalization, without any useful explanations; indeed, what are these “dense connections”? what are hierarchical features?  This text is merely citing unuseful one-sentence summaries from each paper.

..the sub-band residuals of HR image progressively. - what are these? How “progressively”?

..uses the group convolution to make the image SR network light-weight and efficient. - what is a group convolution? And if it’s so good, why have not others done this earlier?

..explore the feature correlations in spatial or channel dimensions .. - what are these? What does “explore” mean?

..squeezing and exciting to obtain the relationship between feature channels. - it is quite insufficient to simply summarize techniques with such short phrases, which explain nothing.

..builds the long-range dependencies on basis of non-local operations .. - all these pseudo-descriptions note mostly desired objectives, rather than actual techniques

..cross-scale feature correlations.  - what are these?

..to focus on the important spatial information. - once again, an objective, not a method

..dilated convolutions to refine the features. - again, if this was so good, why not done earlier?  At each step here, the paper briefly notes a summary term for the methodology, and says that this improves things, without noting any context, e.g., why did such not occur much earlier?  Each successive sentence introduces yet another, separate (independent?) idea, with no contextual discussion of all these ideas.

..measuring the importance of every neuron in a multi-grained way. - how does this work?  Rather than listing a series of separate acronyms, I would have greatly preferred a serious comparison of a select group.

..Although, this method of .. ->

..Although this method of ..

(Also, not good to start a new paragraph by referring to only one of many methods in the previous paragraph)

..Figure 1. Where, Ilr and Isr represent .. ->

..Figure 1, where Ilr and Isr represent ..

Fig. 1 is a nice diagram.

..where, HSF () denotes ..

..where HSF () denotes..

(Never put a comma right after “where”; this annoying item occurs after every equation)

..but also perform .. ->

..but also performs ..

..learning the local residuals of the input. - of what use is this?

..in m-th NRG. 

..in the m-th NRG. 

..bypasses plenty of redundant low-frequency parts .. - there is no discussion of the relevance of frequency; so…why “bypass” here?

..encourages the network to learn more residual information.  - there is no discussion about what is “residual information.”

..compute the attention weights by capturing the cross-dimensional interaction by using a two-branch module. - again, this needs more explanation: 1) what is this “cross-dimensional interaction”? 2) how does using a two-branch module help? 3) how do these two items help to compute the attention weights?

..create interactions between the channel dimension C and the width dimension W. - I have difficulty to understand what these “interactions” are

..of last step to obtain ..->

 ..of the last step to obtain ..

..After exchanging the dimension order .. - rephrase; this sentence lacks a main clause

..generated by following operations.

..generated by the following operations.

..output of triplet attention module.

..output of the triplet attention module.

..the parameter of RTAN is 9.6M, ..

..the number of parameters of RTAN is 9.6M, ..

..to YCbCr space. - what is this?

..about 11days and 10days ..

..about 11 days and 10 days ..

..the PSNR from Ra to Rc have a ..

..the PSNR from Ra to Rc has a ..

..SR performance and make the ..

..SR performance and makes the ..

..the effect of spatial attention block ..

..the effect of the spatial attention block ..

..slightly improves the performance from 32.20 dB to 32.21dB.  - this is a ridiculously small amount

..achieve 0.01dB and 0.03 dB PSNR gains, .. - these too are tiny effects

..structure of the lattices and green lines.  - what is this? Explain more

..Similarly, the parameters of RTAN .. .

.Similarly, the number of parameters of RTAN ..

.., this comes performance degradation.

.., this comes with performance degradation.

..Contrary, the proposed RTAN ..

..However, the proposed RTAN ..

Why cite city locations for some, not all, conferences?

Reviewer 3 Report

In this manuscript, the authors state that in order to effectively utilize the advantages of CNNs and to consider the correlation between the different dimensions of the characteristics, a residual triplet attention module (RTAM) that captures the interactions and interdependencies between the different dimensions of features in the intermediate layers using a small number of parameters is proposed.

This manuscript has shortcomings since the objective of the work is not clearly stated.

Authors should analyze key references [1,2] and state what they bring to the table in relation to them.

  1. Ooi, Y.K.; Ibrahim, H. Deep Learning Algorithms for Single Image Super-Resolution: A Systematic Review. Electronics202110, 867. 
  2. W. Yang, et al. "Deep Learning for Single Image Super-Resolution: A Brief Review," in IEEE Transactions on Multimedia, vol. 21, no. 12, pp. 3106-3121, Dec. 2019.

They should also open the perspectives and cite in the introduction the following references [3,4] on CNN applications in Face Recognition and Medical Imaging.

  1. Adjabi, I. ; et al. Multi-block color-binarized statistical images for single-sample face recognition. Sensors202121, 728.
  2. Ouahabi, A.; Taleb-Ahmed, A. Deep learning for real-time semantic segmentation: Application in ultrasound imaging. Pattern Recognit. Lett.2021144, 27–34.

- Instead of using SSIM, it is better to use the average called MSSIM.

- Why don't authors use other metrics like the Accuray?

- Figures 7 and 8 are not readable and the color code is not defined.

- To validate their approach in the real world, it is imperative to use more realistic image degradations.

Round 2

Reviewer 1 Report

No further changes are required.

Author Response

Thank you very much again for your thorough review and valuable suggestions on our paper writing. These comments and suggestions have helped us to improve the quality of this work.

Reviewer 2 Report

Line 113: the modified residual modules consists .. ->

the modified residual module consists 

193: ..L denotes the number of RTAN layer. 

L denotes the number of RTAN layers. 

195: The mathematical expression as follows. 

The mathematical expression is as follows. 

242: from LR to HR. It bypasses 

from LR to HR, it bypasses 

  • 5.4. Comparison with state-of-the-arts
  • 5.4. Comparison with state-of-the-art

Reviewer 3 Report

There are improvements in the manuscript, but some grey areas remain.

In expression (3), the symbol "*" is not defined. This symbol usually indicates the convolution product and not the (simple) product.
Relation (4) is not clearly written.
The use of the notation delta and delta' in relations (15) and (19) is not universal because delta has a specific meaning in signal and image processing. They should simply be replaced by SIGMO and RELU respectively.
Figures 7 and 8 should be enlarged as they are not readable.
